# Well-Being, Resilience and Social Support of Athletes with Disabilities: A Systematic Review

**DOI:** 10.3390/bs13050389

**Published:** 2023-05-08

**Authors:** Tânia Mira, Aldo M. Costa, Miguel Jacinto, Susana Diz, Diogo Monteiro, Filipe Rodrigues, Rui Matos, Raúl Antunes

**Affiliations:** 1Department of Sport Sciences, University of Beira Interior, 6201-001 Covilhã, Portugal; taniaslmira@gmail.com (T.M.);; 2ISCE—Higher Institute of Lisbon and Vale do Tejo, 2620-379 Ramada, Portugal; 3Research Centre in Sports Sciences, Health Sciences and Human Developmental (CIDESD), 5001-801 Vila Real, Portugal; 4ESECS—Polytechnic of Leiria, 2411-901 Leiria, Portugal; 5Life Quality Research Centre (CIEQV), 2040-413 Rio Maior, Portugal; 6Faculty of Sport Sciences and Physical Education, University of Coimbra, 3040-248 Coimbra, Portugal; 7Center for Innovative Care and Health Technology (ciTechCare), Polytechnic of Leiria, 2411-901 Leiria, Portugal

**Keywords:** adapted sport, disabilities, inclusion, well-being, social support, resilience

## Abstract

Sport for people with disabilities appears to play a positive role in the well-being, resilience and social support of athletes with disabilities. Thus, this systematic review aims to evaluate the effect of adapted sport on the well-being, resilience and social support in a population with disabilities. The Pubmed, Web of Science, Scopus and SportDiscus databases were used, with several descriptors and Boolean operators. A total of 287 studies were identified through searching the databases. After the data extraction process, twenty-seven studies were included for analysis. In general, these studies show that adapted sport has a positive impact on the levels of well-being, resilience and social support resources for people with disabilities, contributing to their personal development, quality of life and integration into society. Considering the impact on the variables studied, these results are important to support and encourage the development of adapted sport.

## 1. Introduction

Although there are barriers to practice (including the challenge of inclusion, limited programs, the inaccessibility of facilities, difficulty in transport and even a lack of social support in various aspects [1,2]), sport, as a tool that promotes health, quality of life and social integration, presents itself as a benefit for people with disabilities, namely in self-confidence [3,4,5], satisfaction and quality of life, self-esteem [3,4,5,6,7], a reduction in suicidal tendencies, a more independent attitude and the motivation to continue evolving [3,4].

On the other hand, the Paralympic Movement has been considered a platform for presenting the abilities of people with disabilities while serving as a catalyst for their rights, ensuring integration, equal opportunities and accessibility [3]. Thus, in addition to issues related to the performance of participants, attention has also been dedicated to understanding the processes underlying the measurement of happiness, leading to well-being being the object of study in the most diverse areas and populations [8,9]. Well-being has been studied in two streams, one according to happiness (subjective well-being representing hedonic well-being) and another according to human potential (psychological well-being representing eudaimonic well-being) [10,11,12,13,14]. Eudaimonia represents living life according to one’s own potential or internal virtue [13]; it derives from personal activities that promote self-realization through the realization of personal potentialities and the achievement of one’s own life goals [15,16]. Hedonism represents life in pursuit of pleasure, what a person feels about his or her own life [13], the satisfaction of desires, the pursuit of pleasure and the rejection of pain [12]. It is characterised by pleasurable experiences, enjoyment, the alleviation of suffering and the achievement of satisfaction [17].

In the literature, hedonic well-being represents subjective well-being, a subjective experience of well-being [15]. Several instruments have been created to study hedonia, with self-report measures of the respective subjective experiences associated with happiness. Subjective well-being, of hedonic premise and complex and multifaceted in nature, evaluates life cognitively and affectively, being subdivided into three components: positive affect, negative affect and life satisfaction [18,19,20].

Several studies seem to prove the positive relationship between practising sports and the increase in well-being [10,21,22,23], including subjective well-being [24,25,26,27,28]. People with motor disabilities who practice sport have higher life satisfaction compared to people with motor disabilities who do not practice sport [3,4,29].

Another topic considered increasingly important in studying people with disabilities is resilience. Fletcher and Sarkar [30] present resilience as “The role of mental and behavioural processes in promoting personal assets and protecting the individual from the potential negative effect of stress” (p. 675). Whether a person reacts to adversity in a positive way depends on the adversities to which they have been subjected and on their own adaptation to them [31]. The study of resilience focused initially on children and more recently on adults [30,32], families and communities that have been exposed in some way to stressful situations, such as the loss of a family member, terrorism, severe illness and natural disasters [30]. Positive findings, although not directly related to people with disabilities, are of extreme importance as these parents and family members are active participants in the education, rehabilitation and training process of their children and family members with disabilities [33,34,35].

Cohn, Fredrickson, Brown, Mikels and Conway [36] studied the relationship between positive emotions and resilience. It was identified that positive emotions lead to higher levels of resilience in the future and, on the other hand, resilience also achieves its effects, in part, through the generation of positive emotion. In stressful situations, people with higher resilience experience more positive emotions than those who are less resilient [36]. Ryff [37] states that well-being sometimes results from actively grappling with adversity. Experiences such as experiences with obstacles, failure and disappointment are necessary to find internal strengths and renew resources that allow one, at the same time, to become aware of one’s own limitations and vulnerabilities [37].

With the increasing development and importance of the concept of resilience, much recent research has emerged within sport [31,38,39,40,41,42,43,44,45,46]. The sports participation of people with disabilities has also shown implications of resilience, especially regarding access to social support, opportunities and meaningful social experiences, in people facing traumatic injuries [47].

Another of the important and indispensable themes in the study of people with disabilities in sport is social support. The study of social support has been the object of interest of several researchers, in the sense of a definition of the conceptual model [48]. A recent approach proposes a model that highlights two life contexts through which people can prosper (successfully facing the adversities of daily life and constantly searching for opportunities for growth and development). Two functions of relational support, fundamental to the experience of development in each context, have been proposed, identifying mediators through which relational support is likely to have long-term effects on prosperity [49]. Social support has been considered as a positive influence in the sport context for this population [50], contributing to improvements in and help with psychological issues, to allow for the development of their abilities and for them to experience and evaluate their limits, using them positively as resources and qualities to accept their difficulties [51].

Sherida, Coffee and Lavalle [52] conducted a systematic review focusing on social support and concluded that coaches, parents and peers have a fundamental role in sport (especially in youth sport) through their positive influence on various factors. They also realized that it is important to adjust the athlete’s support pattern throughout their career as needs change.

To the best of our knowledge, no study has systematically reviewed the literature on the factors associated with the impact of adapted sport on these variables. Focusing solely on people with disabilities, the purpose of the present systematic review was to identify and assess the peer-reviewed scientific literature on the relationship between sport practice and (i) well-being; (ii) resilience; and (iii) social support.

## 2. Materials and Methods

### 2.1. Eligibility Criteria

The present systematic review was performed according to the PRISMA protocol [53,54] and the methods suggested by Bento [55]. The protocol was registered in PROSPERO, with the following number: CRD42022362330 (https://www.crd.york.ac.uk/prospero/export_details_pdf.php, accessed on 13 December 2022).

The PICOS strategy [56,57] was defined as follows: (i) “P” (Patients) corresponded to participants with any type of disability, of any age, gender, ethnicity or race; (ii) “I” (Intervention) corresponded to a sports program, implemented in the population mentioned before, regardless of the intervention period; (iii) “C” (Comparison) corresponded to the comparison between those practicing and those not practicing sports or pre post-intervention program; (iv) “O” (Outcome) corresponded to levels of well-being, resilience and social support as the primary or secondary variables of focus; (v) “S” (Study design) corresponded to intervention studies, randomized controlled trials (RCTs) or not (non-RCTs) and cross-sectional studies.

### 2.2. Information Sources and Research Strategies

The systematic search for articles was conducted between September and 19 December 2022, in four electronic databases: PubMed (all fields), Web of Science, Scopus and SPORTDiscus (title, abstract and keywords). We searched all studies published in English until 31 December 2022.

The reference lists of all selected articles were independently screened to identify additional studies missed in the initial search. The following indexed search descriptors were used across all databases in the following formats: “cerebral palsy” OR “motor disability” OR “motor disorder” OR “physical disability” OR “vision impairment” OR “visual impairment” OR “vision disability” OR “vision disorders” OR “intellectual disability” OR “mental retardation” OR “intellectual disabilities” OR “intellectual developmental disorder” OR “intellectual impairment” OR “hearing impairment” OR “hearing disability” or “hearing loss” OR “multiple disabilities” OR “para athletes” OR “para-athlete” OR “Paralympian” OR “Paralympians” OR “paralympic athletes”) AND sport* AND (“social support” or “social influence” OR “well-being” OR “resilience” OR “resiliency”.

### 2.3. Inclusion Criteria

To be included in this systematic review, studies had to meet the following criteria: (i) articles published in English by 31 December 2022, regardless of country; (ii) intervention studies, RCTs and non-RCTs and cross-sectional studies; (iii) intervention studies with a sports programme; (iv) individuals with disabilities, of the most varied types; and (v) studies with individuals of any age group, gender, race or ethnicity.

### 2.4. Exclusion Criteria

For this review, the following exclusion criteria were considered: (i) studies published before 2001; (ii) studies with participants with other pathologies (e.g., mental illnesses, degenerative diseases); (iii) studies that did not describe the intervention protocol; (iv) studies in which the intervention was not only a sports programme (example: physical exercise + nutrition).

### 2.5. Data Extraction Process

The search was carried out independently by two researchers. In the first phase, all searched manuscript bibliographies were organised using the EndNote software and so duplicated studies were eliminated. Subsequently, the studies were analysed and selected based on the defined inclusion and exclusion criteria. Subsequent to the completion of this process, the results were compared by the researchers.

After reading the full text of all the selected studies, according to the previously defined eligibility criteria, one of the researchers identified the most relevant information published in each study and entered it into a preliminary characterization table (authorship, country reference, objectives, participants, modality, assessment techniques, results and quality score).

## 3. Results and Discussion

### 3.1. Selection of Studies

The initial search conducted in the four databases revealed a total number of 287 identified articles. In the first phase, and after reading the titles and abstracts, 81 were removed for being duplicates and of the 206, 165 were excluded after applying the inclusion and exclusion criteria previously defined for this systematic review. This resulted in 41 articles, of which 2 we were not able to access. Of the 39 articles assessed for eligibility, after a reading of the articles, a sample of 27 studies was considered for analysis, as represented in the PRISMA flowchart (Figure 1).

### 3.2. Quality of the Information

Two reviewers independently evaluated the quality of the studies based on the following domains from recommendations. The methodological quality of the studies was assessed based on Downs and Black’s assessment [58], with the following quality levels: excellent (26–28); good (20–25); fair (15–19); and poor (≤14). No studies were excluded due to low quality scores.

The full text was critically read to confirm that the study fulfilled the inclusion criteria. Twelve studies were excluded, which resulted in a final sample of 27 scientific studies that formed the basis of the present study, as shown in the following Table 1.

All studies presented followed a cross-sectional methodology. By analysing the table, we can see that nine studies were classified as “poor”, seven studies as “fair” and twelve studies as “good”. No article was classified as “excellence”.

### 3.3. Origin

Through the systematic review process, we identified twenty-seven studies: twelve from Europe (Italy [66,79], Ireland [68], UK [59,63,78,80], Poland [7], Portugal [74], Turkey [65] and Ukraine [69,70]), ten from North America (USA [47,60,72,73,81,82,83] and Canada [64,67,75], three from South America (Brazil [61,62] and Chile [77]), one from Asia (Indonesia [76]) and one from Oceania (Australia [71]).

### 3.4. Sports Participants

The samples of the studies reviewed include the following groups of sports: two from wheelchair basketball [36,72], three from swimming [59,66,79], three from wheelchair rugby [47,60,68], one from ice skating [65], one from wheelchair tennis [80] and seventeen from multiple sports [7,26,61,62,63,64,67,69,70,71,73,74,75,76,77,78,82]. At least nine samples are from Paralympic athletes.

### 3.5. Evaluation Protocols/Instruments/Techniques

The studies analysed used different instruments for the same variables: interviews, questionnaires, validated scales and adapted scales.

The main objective of this study was to identify and understand what has been studied about well-being, resilience and social support in disabled athletes practicing a sport modality. The results confirm that these variables have been the focus of researchers’ attention, especially in the last decade.

Of the three variables, we notice that well-being is the most studied [7,59,60,63,64,65,66,67,68,69,70,71,72,73,74,76,79,80,81,82], either when it is analysed individually or in combination with other variables.

Our research resulted in 21 studies that studied well-being, 9 studies that studied resilience and 12 studies that studied social support in athletes with disabilities. Together, well-being and social support were studied in eight studies, well-being and resilience in six studies, resilience and social support in three studies and only two studies studied the three variables together.

### 3.6. Well-Being

All studies that studied well-being revealed positive perceptions between the sport practice of disabled people and this variable. In the analysed articles, well-being was studied in several dimensions and several domains were evaluated, such as well-being [60,64], emotional well-being [66], occupational well-being and spiritual well-being [66], psychological well-being [59,68,70,80], subjective well-being [7,71,72,73,74,82], physical well-being [79] and quality of life [60,63,81].

In the study of the perception of well-being in children and young people with physical disabilities practicing sport, positive perceptions were observed in all domains, including the children with disabilities showing higher values of perception of well-being than their parents. These results are extremely important and should be taken into account, as it is the parents who make decisions regarding their children’s practices, and it is very important to be aware of this detail [81].

In terms of psychological well-being, the positive impact of practicing sport on athletes with disabilities is consistent [68,69,70,76,79,80]. The perception of psychological well-being seems to have a negative correlation with increasing age and persons with congenital disabilities showed higher well-being scores compared to persons with an acquired disability. This may be due to a greater difficulty in adapting to and accepting their own disability [79]. However, in these cases, this difficulty was mitigated by sport in competition, as in the Paralympic group the scores of athletes with congenital and acquired disabilities were similar [79]. Paralympic athletes showed results in line with the standards of the general population. They claim that this is due to sport opportunities. All components of the psychological well-being of Paralympic athletes are significantly higher than students with disabilities who do not participate in sport [69]. Psychological strength is the most important factor supporting self-realization. On the other hand, they determined that two of the four indicators of the self-realization of these athletes are significantly related to the indicators of psychological well-being, purpose in life and meaning in life [70].

Considering the importance and focus of our study on subjective well-being, it was important to note the positive impact of this variable in the various studies [7,71,72,73,82].

Adapted sport presents itself as an enhancer of well-being in athletes with physical disabilities. In studies that analysed subjective well-being, athletes expressed high positive affect and low negative affect [74,81], as well as a significant relationship between positive affect and a strong relationship with peers [81]. High levels of life satisfaction have been associated with playing sports and the experiences provided [7,72,73,74]. These results reinforce the importance that the practice of sports seems to have on the perception of subjective well-being, both in its cognitive dimension (satisfaction with life) and its emotional dimension (positive and negative affect) and are in line with what has been reported in the literature [24,25].

Although the levels of well-being were perceived to be gradually positive as the athletes increased their level of competition, they also concluded that they had well-being needs, suggesting an interaction between physical pain, emotional regulation, lack of purpose outside of sport and lack of self-acceptance. Others reported negative emotions, frustration, bitterness and uncertainty, which may have to do with the fact that they are elite athletes who feel poorly supported and afraid of losing their careers because they do not have the financial capacity to train as they would like to. If these athletes reported emotional unbalance as a need in terms of well-being at the end of a major competition, such as the Paralympic Games, including feelings of loss, lack of guidance and depression, it was also perceived that when strong well-being was manifested, it was associated with personal growth, optimism, strong social support networks and contribution to various communities. Personal growth and optimism have been related, as a result of life satisfaction, with the ability to deal with what life provides in a positive way [71].

These results support our belief in the importance of adapted sport in contributing to well-being in people with disabilities.

The most common limitations of the studies analysed in this systematic review are that they include a restricted group, such as participants from a single sport or athletes from a single country or state [59,63,72,82,83], and no experimental studies were found for the well-being variable. It is suggested that in future investigations, comparisons of people with and without motor disabilities, other types of disabilities, other sports, and analysis of the existing differences as well as experimental studies should be considered [83], to evaluate the impact of sports practice on the variables.

### 3.7. Resilience

Satisfaction with life has been studied with other variables such as resilience, with a positive relationship observed between the two variables. In the analysed studies, it is a consistent conclusion that athletes with higher levels of resilience have higher life satisfaction [7,72,73]. Resilience is mentioned as a significant predictor of life satisfaction and sport involvement [73].

Athletes with disabilities who play sports have demonstrated high levels of resilience qualities: life satisfaction, optimism, resilience and social focus [7].

Resilience has been widely studied as a quality present in people with disabilities in the sense that they are people with resources that allow them to protect themselves and thus overcome the adverse effects of exposure to risk [84] and, in the case of this systematic review, this includes practicing sport [7,47,60,61,72,73,74,77,78].

Individuals with spinal cord injury and myelomeningocele (spina bifida) showed higher levels of resilience compared to the lower resilience observed in people with cerebral palsy. This result can be explained by the functionality existing in each disability. Thus, athletes who presented high levels of resilience showed higher levels of quality of life [61]. Higher levels of resilience were observed in people with disabilities and correlated especially with their own relationship with their body and sexual esteem. They concluded that sport has the ability to improve athletic identity and self-esteem, causing people with disabilities who participate in sport to develop self-confidence to face and cope with their own disability and its challenges [77]. The development of resilience in athletes with spinal cord injury is a multifactorial process that involves pre-existing factors and experiences, unpleasant emotions, various types and sources of social support that generate opportunities and special experiences that lead to different cognitive and behavioural strategies and, consequently, motivation to adapt to change [47,71].

We observed that paralympic athletes, as a rule, are conscious that their ability to overcome difficulties is related to their physical and mental experiences and challenges related to disability. They consider that the development of these resilient characteristics and ability to cope with physical and emotional pain comes from the constant exposure to pain and stress experienced [78]. The wheelchair tennis athletes who participated in the study by Richardson et al. [80] highlighted an important detail. Participation in this sport strengthens one’s resilience, which then transfers off the court. By overcoming challenges as players, it allows them to develop resilience which they can apply in their daily lives.

Like the well-being variable, we noticed that no experimental study on resilience was found in the present review.

In the studies reviewed, we found some limitations of the study of resilience, such as the different ways and different moments of resilience assessment between studies, resilience not being measured in relation to an adverse event and some researchers see limited value in assessing resilience in the absence of an adverse situation [72,73]. Additionally, the condition of the athlete’s disability, congenital or acquired, may influence resilience levels [47]. Samples with multiple sports included may make it difficult to generalise findings due to the nature of and differences between the specific sports [78].

Future studies may assess resilience by type of disability, congenital or acquired, and, when acquired, by the relationship with the time it was acquired, as it may be different depending on the condition and context.

### 3.8. Social Support

We perceived that self-esteem and social support have a reciprocal relationship, low self-esteem is associated with not seeking support and a greater tendency for misconduct. Involvement in sport for people with intellectual disabilities has added value by acting as an intervention to increase resilience [63]. Athletes who have higher levels of resilience and social support show more commitment to their sport [61].

There is a consensual agreement on the importance of social support, mainly from family, friends and coaches [47,66,74,75,83]. In this systematic review, the first study to address the issue of social support was prepared by Swanson and Zhao [83]. They reported that social support is essential to build teamwork, which provides support in the various challenges and tasks that are inherent and adjusted to the characteristics and needs of the athlete so that they can be successful [83]. Social support is identified as one of the most important factors in dealing with challenges and recovering from adversity; it is fundamental to the process of developing resilience. The sources of social support are multiple, from family, therapists, colleagues and coaches, among others [47].

Sport experiences provide an improvement in social skills, which in turn is related to well-being and social support. The efforts and reinforcing approval they receive from coaches, parents and peers provides athletes with a sense of social acceptance [66,74].

Athletes who experienced higher levels of social support inside and outside of sport competitions showed higher levels of engagement with their sport [60].

Recently, Aichtinson et al. [59] developed the model “The Podium Illusion”, where they identified twenty-five forms of support mentioned by athletes with disabilities. Of the various supports mentioned, they highlighted the importance of the vital role of “staff” in enhancing their training and performance, including coaches, psychologists, physiologists, nutritionists, biomechanists, etc. Family, friends and other performance agents were associated with the necessary and indispensable support by the provision of mental health care and overall happiness. This provided assurance of well-being allowed a perception of improvement in the athlete’s performance. The coach was valued as an inspirational figure of motivation, providing improvements in levels of self-esteem and well-being, resulting in improved overall performance. The strong relationship with the coach is extremely important for these athletes [59].

The results found in the analysed studies are significantly important. There is a consensus that social support from family, friends and coaches has a decisive impact on the sport practice of athletes with disabilities [59,60,63,75,76,83]. It is suggested that it is beneficial to analyse the coaches’ perception of the social support of their athletes [75].

One of the limitations in studying social support is the diversity of assessment tools, given the various components and characteristics of social support. Future studies should focus on only one instrument or should carry out subgroup analyses. The timing of the assessment of social support can be considered a limitation [75]. As with the study of resilience, the athlete’s disability status, congenital or acquired, may influence levels of social support [78]. At the same time, future studies may assess social support by type of disability, congenital or acquired, and when acquired, by the relationship with the time it was acquired and the respective needs.

Furthermore, it seems pertinent that future studies focus on the relationships that can be established between these all variables, particularly regarding adapted-sports athletes (for example, the analysis of the effect that social support and resilience can have on the well-being of athletes).

## 4. Conclusions

From the analysis of the studies included in this systematic review, we can conclude that sport seems to have a positive role in the well-being of athletes with disabilities in the various domains.

All studies that studied well-being revealed positive perceptions between the sport practice of disabled people and this variable.

At the same time, these studies confirm and strengthen the relationship between life satisfaction and resilience, in the sense that athletes with higher levels of resilience have greater life satisfaction. Consequently, they also present greater involvement and commitment to their sport.

We can observe that there are several studies that strengthen the importance of the practice of sports as a significant social opportunity that helps in the resilience process of people who face adverse situations derived from their own disability.

We also concluded that social support for these athletes is extremely relevant for the improvement of their career. Family, friends and other social agents were associated with indispensable support for the possibility of sports practice. These results confirm that the participation of educated and trained people to support the development of these athletes’ careers will consequently have positive results.

We consider these results important to support and encourage the development of adapted sport, which should be taken into consideration by policymakers and other organizations due to the positive impact that it has on contributing to levels of well-being, resilience and social support resources in people with disabilities, contributing to improvements in their personal development, their quality of life and their integration into society.

## Figures and Tables

**Figure 1 behavsci-13-00389-f001:**
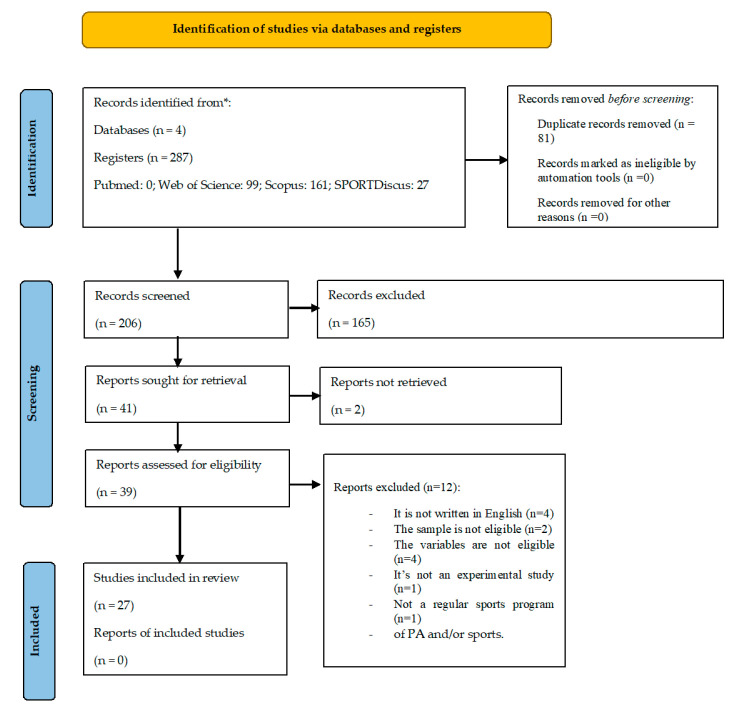
PRISMA flow diagram illustrating each phase of the search and selection process.

**Table 1 behavsci-13-00389-t001:** Characteristics of the studies.

Author, Reference, Country	Aims	Participants	Modalities	Evaluation Techniques	Results	Quality Score
**Aitchison et al.** [59] **United Kingdom**	Explore the experiences of social support in elite British para-swimmers and the influence on their wellbeing and performance	*N* = 9 British para-swimmers (3 male, 5 female, mean age 24.9 years). Disabilities: physical, visual and intellectual disabilities	Swimming	Semistructured interviews with duration ranging from 48 to 88 min (mean ± SD length 61 ± 15.3 min)	Social support in British para-swimmers is important for performance and wellbeing, especially a strong coach–athlete relationship, motivational and supportive teammates.	Fair
**Atkinson and Martin** [60] **U.S.A.**	Understanding if grit, hardiness, resilience and the added variable of athlete social support would predict life satisfaction and sport engagement in wheelchair rugby athletes	*N* = 87 adult athletes (80 males and 7 females). 35 in-person and 52 online. M age = 35.94 years, SD = 9.26) Disabilities: amputation (*n* = 6), spinal cord injury (*n* = 60), cerebral palsy (*n* = 9), other (*n* = 2).	Wheelchair rugby	Connor–Davidson Resilience Scales revised Norwegian Hardiness scale Satisfaction with Life Scale. 16-item Athlete Engagement Questionnaire 22-item Athletes’ Received Social Support Questionnaire	The resilience predicted 32% of the variance in life satisfaction and was the only significant predictor of life satisfaction. Grit, social support, resilience and hardiness were all significant predictors.	Good
**Cardoso and Sacomori** [61] **Brazil**	Examine resilience in the specific subgroupof Brazilian competitive athletes with physical disabilities. Test the validity of the Wagnild and Young (1993) Resilience Scale in Brazil for this population	*N* = 136 athletes Aged 18 or over with Disabilities: physical	Track and field, table tennis, swimming, weightlifting, basketball, rowing and tennis.	Semistructured interviews The questionnaire consists of 25 items	Participants were observed to have achieved higher resilience scores, while those with cerebral palsy obtained the lowest and those with amputations or polio obtained intermediate scores. The participants in this research study displayed a significantly lower mean resilience than those reported in other studies with the general population.	Fair
**Cardoso et al.** [62] **Brazil**	Describe the importance of structural and human resources support for Brazilian Paralympic athletes	*N* = 10 Paralympic athletes Athletics (7 males and 3 females) M age 28 ± 7.95. Swimming *N* = 10 (8 male and 2 female) M age = 22.50 ± 4.54 Disabilities: physical	Athletics and swimming	Semistructured interview	The results show that structural support and human resources support are considered fundamental by the interviewed athletes.	Poor
**Crawford, Burns and Fernie** [63] **United Kingdom**	Investigate the relationship between psychological resilience and vulnerability factors and involvement in the SO compared to being involved in sport not through the SO, and no sports activity	*N* = 101 (44 female and 57 male) M age 35.1 (range 18–67) Disabilities: intellectual	Athletics, football, judo, swimming, bowling and multiple sports	Demographic questionnaire Wechsler Abbreviated Scale of Intelligence (WASI) Rosenberg Self-esteem Scale The Life Stress Inventory Social Support Self Report	The results indicate that there is an association between involvement in the SO and reduced stress, increased quality of life and higher self-esteem. The hypothesis of increased social networks was not demonstrated. The findings provide further evidence of a positive association between sport involvement and increased psychological wellbeing, especially for those involved in the SO.	Good
**Downie and Koestner** [64] **Canada**	Study 1 compared the relation between success in the Paralympics versus the Olympics and national subjective well-being and life expectancy. Study 2 conceptually replicated these results using the standings of national men’s and women’s soccer teams	Olympics and Paralympics athletes	Olympics and Paralympics modalities (not specified) Soccer	Olympic and Paralympic Performance. Happiness. Data were taken from the World Database of Happiness	Study 1—There were mean differences in happiness for those countries that had a Paralympics team compared to those that did not. This suggests that even though countries without a Paralympics team were no less wealthy, or happy, their physical well-being was reduced. Study 2—Countries with a female soccer team were happier.	Poor
**Dursun et al.** [65] **Turkey**	Assess the effects of ice skating on the psychological well-being, self-concept and sleep quality of children with hearing or visual impairments	*N* = 40 students (20 visually impaired and 20 hearing impaired) Aged 8–16 Disabilities: visual and deaf	Ice skating	A self-report form of the Strengths and Difficulties Questionnaire (SDQ) The Piers–Harris Children’s Self-Concept Scale (PHCSCS) Pittsburgh Sleep Quality Index (PSQI)	There was a significant improvement in self-concept, behavioural and emotional problems and sleep quality (*p* < 0.05 for each) in of the children with hearing impairment. Although the sleep quality (*p* = 0.019) and emotional problem scores (p ¼ 0.000) of the visually impaired children improved, self-concept, peer relations and hyperactivity scores of these children worsened (p50.05 for each).	Fair
**Fiorilli et al.** [66] **Italia**	To examine the perception of well-being, social integration and emotional problems of Down Syndrome (DS) subjects and to investigate whether parents and their children with Down Syndrome have the same opinion on the problems caused by DS	*N* = 93 participants with DS 58 swimmers (aged 16.31 ± 1.55), 35 DS sedentary subjects (aged 16.06 ± 1.39), and their parents (*n* = 93). Disabilities: Down Syndrome	Swimming	2 questionnaires (SDQ) were individually administered: self-reported version (SDQ-SR), completed by the DS participants, and the parental version (SDQ-P), completed by their parents	Results showed significant differences between sportive vs. non-sportive groups in the overall domain scores, with better results for the sportive group. Parents of DS non-sportive participants underestimated their children’s problems in 6 of the 8 domains. The results validate the hypothesis that foresees a positive relation between well-being perception and sport activity and competitions in individuals with MR, such as DS.	Good
**Hamdani et al.** [67] **Canada**	To understand the state of perceived wellness and wellness-promoting behaviours of children and youth with IDD from multi-stakeholder perspectives	*N* = 35 in-person surveys of athletes *N* = 352 online surveys of caregivers (*n* = 240) and coaches (*n* = 112) Disabilities: intellectual	Special Olympics	A cross-sectional Likert survey methodology	Athletes, caregivers and coaches generally agreed rather than disagreed with wellness statements, with the exception of coaches’ responses regarding healthy nutrition. Athletes agreed more than caregivers and coaches that they engaged in some wellness-promoting behaviours (i.e., calming oneself down, participating in their communities).	Fair
**Haslett, Fitzpatrick and Breslin** [68] **Ireland**	To interpret participation in wheelchair rugby through the conceptual lens of the SRM	*N* = 10 male athletes from three clubs (M age = 33.1 years, age range: 22–53 years).	Wheelchair Rugby	A semistructured interview guide was developed based on the components of the SRM	The results indicate that in disability sport participation, the experience of social oppression, inequality and cultural stereotypes of disability can be synonymous with the personal experience of physical impairment.	Poor
**Kokun et al.** [69] **Ukraine**	To determine the influence of sports on Paralympic athletes’ personal development	*N* = 106 Paralympic and Defilimnical (Age 16–53, 84 men and 22 women) *N* = 191 students (Age 17–25, 91 without health problems, 98 with disabilities not engaged in sports) Disabilities: physical and deaf	Paralympic sports	The Ryff Scales of Psychological Well-Being and S. Maddi’s Personal Hardiness test.	Paralympic athletes achieve an optimal level of psychological well-being and a significant increase in all psychological hardiness components.	Fair
**Kokun, Serdiuk and Shamyco** [70] **Ukraine**	To investigate the personal characteristics supporting Paralympic athletes’ self-realization in sports.	*N* = 106 Paralympic and Defilimnical in different sports (Age 16–53, 84 men and 22 women) Disabilities: physical and deaf	Paralympic sports (football, fencing, powerlifting, sitting volleyball, judo, canoeing in pairs, swimming)	Self-efficacy scale of R. Schwarzer and M. Yerusalem; Ryff’s Scales of Psychological Well-Being; S. Maddi’s Personal Hardiness test (adapted by D. Leont’ev); the modified techniques of scaled self-estimation; The Self-Organization Questionnaire	The most important personal characteristic supporting Paralympic athletes’ self-realization in sports is their psychological hardiness, since all four of its indicators have sufficiently close correlations with three of the four self-realization indicators—“Satisfaction with own sports career”, “Fastness of sportive goal setting after achievement of a previous one” and “Reaching of top-achievements in sports”. Paralympic athletes’ rapidity in sportive goal setting after achievement of a previous one is significantly related to all ten psychological well-being scales, and their clarity of perception of own future in sports correlates with five scales.	Fair
**Macdougall et al.** [71] **Australia**	Explore well-being in para-athletes in a way that is consistent with theoretical perspectives	*N* = 23 para-athletes (10 female and 13 male) M age = 28.5 years, age range = 16–53 years) Disabilities: physical	Athletics, boccia, canoe slalom, cycling, swimming, table tennis, triathlon, wheelchair basketball, wheelchair rugby	Semistructured interviews	The well-being needs and strengths of para-athletes differed across gender, sport, level of competition and nature of impairment. Well-being strengths were perceived to increase as athletes increased their level of competition, and included personal growth, optimism, strong social support networks and contributing to multiple communities.	Poor
**Machida, Irwin and Fetz** [47] **U.S.A.**	Examine the resilience process of sport participants with acquired spinal cord injury, and the role of sport participation in the resilience process	*N* = 12 men with aged 21 to 41 years Disabilities: physical	Wheelchair rugby	Semistructured interviews	The development of resilience is a multifactorial process involving pre-existing factors and pre-adversity experiences, disturbance/disturbing emotions, various types and sources of social support, special opportunities and experiences, various behavioural and cognitive coping strategies, motivation to adapt to changes and learned attributes or gains from the resilience process.	Poor
**Martin et al.** [72] **U.S.A.**	To predict both general and sport-specific quality of life using measures of grit, hardiness and resilience	*N* = 75 (74 males and 1 female) M age = 37.0 years, SD = 11.01 Disabilities: physical	Wheelchair basketball	8-question Short Grit Scale Connor–Davidson Resilience Scales revised Norwegian hardiness scale (Dispositional Resilience Scale Satisfaction with Life Scale 16-item Athlete Engagement Questionnaire	Hardiness and resilience combined to predict 26% of the variance in life satisfaction, whereas grit was not a significant predictor. Athletes higher in both resilience and hardiness expressed greater life satisfaction compared with athletes lower in hardiness and resilience.	Good
**Martin et al.** [73] **U.S.A.**	Determine if grit, hardiness and resilience predicted life satisfaction and sport engagement in parasport athletes	*N* = 40 (22 ice hockey athletes and 18 wheelchair rugby) (M age = 32.0; SD = 8.6) Disabilities: physical	Ice hockey, wheelchair rugby	Short Grit Scale (Grit-S)10-item Connor–Davidson Resilience Scale (10- item CD-RISC) Norwegian hardiness scale (Dispositional Resilience Scale 15 (DSR-15) Satisfaction with Life Scale (SWLS) 16 item Athlete Engagement Questionnaire (AEQ)	Hardiness and resilience are important predictors of life satisfaction, with grit being irrelevant, and resilience and grit are important predictors of sport engagement. Overall resilience appears to be the most critical predictor for both outcomes across the three studies.	Good
**Mira et al.** [74] **Portugal**	Characterizing subjective well-being, resilience and social influence	*N* = 31 of the 33 athletes of the Portuguese Paralympic team M age = 34.45 ± 11.7 years (21 men and 10 women) Disabilities: physical	Multiple sports	Social support perceived by athletes with disabilities, a scale based on the recommendations of Jago et al. Subjective well-being was assessed through Satisfaction with Life Scale Positive and negative affect were evaluated through PANAS—The Positive and Negative Affect Schedule Brief Resilience Scale (BRS)	Athletes perceive a positive affect superior to negative affect. Regarding social support, the perception of support by the coach is the one with the highest value. The bivariate correlation was observed between life satisfaction and positive affect (medium), between positive affect with social support from parents (high) and between positive affect with social support from friends (medium). Resilience displayed a negative and significant association with the negative affect (high).	Good
**Monton et al.** [75] **Canada**	Explores the role of sport–life balance and well-being on athletic performance.	*N* = 47 Olympic athletes *N* = 14 Paralympic athletes *N* = 11 not specified (2 male, 40 female and 10 athletes that did not specify) Disabilities: physical	Multiple sports	Mixed-methods research design consisting of an online survey and a series of semistructured, follow-up interviews.	Overwhelmingly, athletes felt they received the most support from family and friends. These results exemplify the importance of athletes’ external support systems.	Poor
**Mudjianto, Widya and Ubad** [76] **Indonesia**	Determine the quality of life of Paralympic athletes who are members in West Java Peparnas Pelatda 2016	Handicapped/disabled athletes from Pelatda West Java	Not specified	Questionnaire WHOQOL-BREF	From the results of the analysis and calculation of the data obtained, the four domains of physical health, psychological well-being, social relationships and the relationship with the environment showed a good average range.	Poor
**Porto, Cardoso and Sacomori** [77] **Chile**	To analyse the association of team sports practice and physical and psychological factors with sexual adjustment in men with paraplegia	*N* = 60 men (30 wheelchair basketball and/or handball athletes and 30 non-athletes) Disabilities: physical	Wheelchair basketball and/or handball athletes	The Resilience Scale	Resilience was correlated with and predictive of sexual adjustment.	Good
**Powell and Myers** [78] **United Kingdom**	Understand the lived experiences of mentally tough Paralympians, aiming to conceptualize MT in a Paralympic context and investigate its development	*N* = 10 Team Great Britain Paralympic athletes (9 male and 1 female M age = 27.9, SD = 7.1) Disabilities: physical	Multiple sports	Semistructured interview	The development of MT requires a series of formative experiences, combined with support and coping resources. Athletes in general would benefit from exposure to highly demanding situations in a supportive environment to develop mentally tough characteristics and behaviours and to develop personalized cognitive strategies.	Poor
**Puce et al.** [79] **Italy**	Investigate the role of competitive sport practice in enhancing the self-perceived psychophysical well-being of some select participants	*N* = 100 Paralympic athletes *N* = 100 affected by impairment who do not practice competitive sport Disabilities: physical	Swimming	Psychological General Well-Being Index and the Short Form indices	Possible positive psychophysical benefits of competitive sport practice for young people affected by physical or intellectual impairment.	Poor
**Richardson et al.** [80] **United Kingdom**	To assess the psychosocial impact of participating in sport at an individual level and the impact of participating in sport in challenging cultural perceptions of disability	*N* = 16 (14 males and 2 females) M age 29 years (age range 18–40) Disabilities: physical	Wheelchair tennis	Semistructured interview	Wheelchair tennis players perceived that their participation in sport enhanced their psychosocial well-being. Three broad themes emerged from analysis of the interviews: (1) developed transferrable skills, (2) perceived personal growth and (3) benefits of an athletic identity.	Good
**Saphiro and Malone** [81] **U.S.A.**	Examined the relationship between athlete and parent perceptions of health-related quality of life (HRQoL) and the relationship between the athletes’ perceived HRQoL and subjective exercise evaluations	*N* = 70 athletes (47 males and 23 females) (M age = 15, SD = 2.92) Disabilities: physical	Swimming (n 5 3), wheelchair basketball (n 5 23), wheelchair handball (n 5 32), and a weekend multisports program (n 5 12	Pediatric Quality of Life Inventory (PedsQL) Subjective Exercise Experiences Scale (SEES)	Athletes with disabilities reported higher perceptions of HRQoL than their parents reported for them on physical, emotional and social functioning subscales with moderate to high effect sizes. Positive well-being subscale from the SEES was significantly related to overall HRQoL and was a significant predictor of overall HRQoL.	Good
**Shapiro and Martin** [82] **U.S.A.**	Examine athletic identity, affect and peer relations of youth athletes with physical disabilities and selected relationships among these variables	*N* = 36 (27 males and 9 females) M age = 16 Disabilities: physical	Swimming, wheelchair basketball, wheelchair handball, multisport program	Private–Public Athletic Identity Scale The Positive and Negative Affect Schedule The Peer Relations Scale	Participants reported stronger private athletic identity individual item scores compared with public athletic identity and expressed high positive affect and low negative affect. They also expressed strong peer relations. A significant relationship between positive affect and peer relations was found.	Good
**Sikorska and Gerc** [7] **Poland**	To describe selected aspects of a good life in Polish Paralympic athletes in light of positive psychology	*N* =30 disabled athletes (M = 26.5 years of age) *N* control group = 30 healthy young adults (M = 25.9 years of age) Disabilities: physical	Skiing, cycling, swimming, fencing, basketball	The You and Your Life Questionnaire The Resilience Scale for Adults by Friborg et al. (RSA) The Satisfaction with Life Scale (SWLS) Personal Values List (LWO)	A statistically relevant difference between the two groups can be identified with regard to resilience, concerning the structured style factor and with regards to courage in the level of the endurance factor. Disabled athletes choose the following as highly assessed values: being useful, courage and firmness.	Fair
**Swason, Colwell and Zhao** [83] **U.S.A.**	This study explored relationships among four sources of motivation and six forms of social support	*N* = 19 wheelchair athletes, (33 male and 60 female) M age = 19.79, SD = 4.93 Disabilities: physical	Wheelchair basketball	Survey	Importance of social support types differed according to skill level, playing level, years played and future playing intentions.	Good

## Data Availability

Not applicable.

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
