# Peer review of "Well-Being, Resilience and Social Support of Athletes with Disabilities: A Systematic Review"

_behavsci, 2023, doi:10.3390/bs13050389_

Round 1
Reviewer 1 Report
Dear authors, thanks for submission your paper for publication. The paper is well prepared, to increase the quality of the study the following recommendations were suggested:
1) Abstract: please specify which methods were used ("methodological process") and add the main contributions of this study.
2) Section 3. "Results" contents the information regarding the used methodology and data extraction process, and do not show the results of the study. This information should be in section 2 "Materials and Methods".
3) Please check the repetitions and make needed corrections as:
page 4, lines 164-165: "Of the 39 articles assessed for eligibility, after a full reading of the articles, a sample of 27 studies was considered for a full analysis."
page 5, lines 175-177: "The full text was critically read to confirm that the study fulfilled the inclusion criteria. This resulted in the exclusion of 12 studies, which resulted in a final sample of 27 scientific studies that formed the basis of the present study, as shown in the following table 1: "
4) Section 4. "Discussion" can be renamed as "Results and Discussion", as it includes the main results of the conducted analysis.
5) Section 4: Please add a conceptual model of the studied factors and their influences, it will greatly enrich the paper.
6) Conclusion: please specify the contributions of the study to existing literature; provide recommendations based on obtained results to policymakers and others; mention the main limitations of this study and provide suggestions for future studies.
Author Response
Response to REVIEWER 1
1) Abstract: please specify which methods were used ("methodological process") and add the main contributions of this study.
Response: dear reviewer, we have substituted the term "methodological process" for data extraction process, which in fact is the correct term. At the same time, we added a sentence with the main contributions of our study.
2) Section 3. "Results" contents the information regarding the used methodology and data extraction process, and do not show the results of the study. This information should be in section 2 "Materials and Methods".
Response: dear reviewer, we appreciate your suggestion. However, our paper was written according to the PRISMA 2020 Guidelines. According to the PRISMA 2020 cheklist, the information you refer to should be in the results section. We attach the reference: Page, M.J.; McKenzie, J.E.; Bossuyt, P.M.; Boutron, I.; Hoffmann, T.C.; Mulrow, C.D.; Shamseer, L.; Tetzlaff, J.M.; Akl, E.A.; Brennan, S.E.; et al. The PRISMA 2020 Statement: An Updated Guideline for Reporting Systematic Reviews. BMJ 2021, 372, n71, doi:10.1136/bmj.n71
3) Please check the repetitions and make needed corrections as:
page 4, lines 164-165: "Of the 39 articles assessed for eligibility, after a full reading of the articles, a sample of 27 studies was considered for a full analysis."
Response: The information has been corrected.
page 5, lines 175-177: "The full text was critically read to confirm that the study fulfilled the inclusion criteria. This resulted in the exclusion of 12 studies, which resulted in a final sample of 27 scientific studies that formed the basis of the present study, as shown in the following table 1: "
Response: The information has been corrected.
4) Section 4. "Discussion" can be renamed as "Results and Discussion", as it includes the main results of the conducted analysis.
Response: Thank you for your suggestion, which has been accepted.
5) Section 4: Please add a conceptual model of the studied factors and their influences, it will greatly enrich the paper.
Response: Dear reviewer, thank you for your suggestions. In the introduction section, we have addressed the variables studied, as well as their conceptual model. To mention them again in this section would duplicate the information. We think that the introduction is clear enough to make the reader aware of the variables studied. In any case, we have included more information.
6) Conclusion: please specify the contributions of the study to existing literature; provide recommendations based on obtained results to policymakers and others; mention the main limitations of this study and provide suggestions for future studies.
Response: Dear reviewer, we think your comment is extremely important to improve the clarity and results of the study. Since we divided the discussion section into 3 parts (one for each variable studied), in the last paragraphs of each we presented the main limitations and future recommendations. Similarly, in the conclusion we have highlighted the real practical implications. Nevertheless, we have reviewed the information mentioned by you.
Reviewer 2 Report
In my opinion, the work meets the requirements for carrying out a systematic review, only a reference to the methodology used for the review could be included in the abstract and the keywords that already appeared in the title could be eliminated. Likewise, the theme is very interesting and provides results that support the positive effect of sport for people with disabilities.
the main aspects that must be modified in the work:
In the introduction:
Review the objective of the study, according to this objective only experimental or quasi-experimental works could be included. Therefore, it would have to be modified by including works that analyze these aspects without studying their cause-effect.
In method section:
In the data extraction process: specify the extracted variables in a table (clearly specifying what modality refers to and using the appropriate term to collect methodological aspects (type of design and research instruments; it would also be interesting to include in those works based on a intervention, the characteristics of itself. On the other hand, it is understood that the limitations are also included in the table since later reference is made to them in the discussion section.
Likewise, make reference to the inclusion of the determination of the quality of the work as an aspect added by the researchers from the evaluation of the works.
Results:
Results section could be very improved. Include results of all the variables analyzed, not only origin, sports modality practiced, and research instruments. The most relevant aspects are missing, such as those related to the research objectives and their results, sample, type of research and characteristics of the intervention.
The discussion should be organized according to the objectives of the work as well as the conclusion.
Author Response
Response to REVIEWER 2
In my opinion, the work meets the requirements for carrying out a systematic review, only a reference to the methodology used for the review could be included in the abstract and the keywords that already appeared in the title could be eliminated. Likewise, the theme is very interesting and provides results that support the positive effect of sport for people with disabilities.
Response: We would like to thank you for the opportunity to submit a revised draft of our manuscript. The comments of the reviewer were of the utmost importance to help clarify and improve our work.
the main aspects that must be modified in the work:
In the introduction:
Review the objective of the study, according to this objective only experimental or quasi-experimental works could be included. Therefore, it would have to be modified by including works that analyze these aspects without studying their cause-effect.
Response: Dear reviewer, thank you very much for the comment. We agree with you and will proceed accordingly.
In method section:
In the data extraction process: specify the extracted variables in a table (clearly specifying what modality refers to and using the appropriate term to collect methodological aspects (type of design and research instruments; it would also be interesting to include in those works based on a intervention, the characteristics of itself. On the other hand, it is understood that the limitations are also included in the table since later reference is made to them in the discussion section.
Response: Dear reviewer, this is a difficult process to perform due to the heterogeneity of information. In our opinion, it would bring little robust and clear results. Since the systematic review is a preliminary study, we prefer to look at it only as a sport practice in general. Future studies should focus on the population of interest, the sport practiced, and other point mentioned in your commentary.
Likewise, make reference to the inclusion of the determination of the quality of the work as an aspect added by the researchers from the evaluation of the works.
Response: Dear reviewer, in point 3.2 (line 177-182), we mention the evaluation process. In the table of extracted variables, we put the result of the evaluation. At the same time, we have reinforced this information in line 189-191.
Results:
Results section could be very improved. Include results of all the variables analyzed, not only origin, sports modality practiced, and research instruments. The most relevant aspects are missing, such as those related to the research objectives and their results, sample, type of research and characteristics of the intervention.
Response: Thank you very much for your comment. We consider that in the previous reply we have already responded to this reviewer's comment. Nevertheless we are available for any further clarification.
The discussion should be organized according to the objectives of the work as well as the conclusion.
Response: Many thanks for the reviewer's comment. Indeed the order of presentation was not consistent with the objectives. Therefore, and taking into account that we consider the discussion to be well organised, we have chosen to reorganise the objectives and the conclusion so that coherence can be maintained.
Author Response
Response to REVIEWER 3
- the introduction needs to be strengthened by providing a succinct and convincing rationale the relationship of well-being, resilience and social support for athletes with disabilities. I suggest that the introduction could be revised to be more focused on the issue, providing clear definition of terms and rationale for the study.
Response: Dear reviewer, our aim was not to analyse the relationship between the variables, but the relationship of sports practice with each of them separately, as we did in the discussion. In this way, the introduction section sought to analyse the existing literature on the conceptual models of the variables analysed, as well as the possible associations with the practice of sport. However, information about future studies was added.
- Much information was provided in the section of introduction. The author could consider present the manuscript in consecutive manner.
Response: Thank you for your comment. As we mentioned before, the introduction tried to analyse the conceptual framework of the different existing variables as well as their relationship with sport practice, especially in people with disabilities. However, we have reorganised a part of the introduction, also taking into account the comments of the different reviewers.
- The last paragraph of introduction seemed to describe the benefits of atheles with disability. I suggest that the paragraph move to in the beginning of introduction.
Response: Thank you for your comment. We proceed according to the suggestion.
- There were to many references in Table 1. The literature reviews are a critical appraisal of a subject and are nor only an academic requirement bur essential when planning a research project and for placing research findings into context.
Response: Dear reviewer, each of the 27 references shown in Table 1 correspond to the articles extracted for analysis.
- In discussion section, the lack of description on high positive affect and low negative affect of athletes.
Response: Thank you very much for your comment. Information on the outcome regarding positive and negative affect has been reinforced, as suggested by the reviewer.
- It is not clear in discussion section. A more in-depth discussion is needed to resilience involved in people with disabilities.
Response: Dear reviewer, as suggested by another reviewer we have added the results section to the discussion. At the point where we talk about the well-being (line 227), resilience (line 282) and social support (line 327) sections, we begin by addressing the results of our systematic review on the various points, comparing with the literature. We end with the limitations presented by the authors of the primary studies, as well as suggestions for future studies. However, we have reformulated the resilience section, according to your suggestion.
- The author should be made more specifically on future studies.
Response: Dear reviewer, thank you very much for your suggestion. The information regarding suggestions for future studies has been added and clarified (line 377-380).
Round 2
Reviewer 1 Report
Dear authors, thanks for your revised paper. The overall structure and quality are significantly improved.